# An Adaptive Three-Dimensional Improved Virtual Force Coverage Algorithm for Nodes in WSN

**Mengjian Zhang** [1,2], **Jing Yang** [1,3,*] and **Tao Qin** [1,3]

1   Electrical Engineering College, Guizhou University, Guiyang 550025, China; mjz960106@163.com (M.Z.); tqin@gzu.edu.cn (T.Q.)
2   School of Computer Science and Engineering, South China University of Technology, Guangzhou 510006, China
3   Guizhou Provincial Key Laboratory of Internet + Intelligent Manufacturing, Guiyang 550025, China
*   Correspondence: jyang7@gzu.edu.cn

**Abstract:** The original virtual force algorithm (VFA) is proposed for the two-dimensional node coverage and localization of wireless sensor networks (WSN). This work proposes a novel three-dimensional improved virtual force coverage (3D-IVFC) algorithm for the 3D coverage of nodes in WSN. Firstly, the node coverage theory is analyzed, which is about node coverage in three-dimensional space. Secondly, an improved three-dimensional space virtual force coverage method is proposed with an adaptive virtual force parameter control strategy. Finally, simulation experiments are utilized to verify the performance of the 3D-IVFC approach. Experimental results show that during random initialization, the average coverage rate of the improved 3D space coverage algorithm was increased by 0.76% and the deployment time was reduced by 0.1712 s; during center initialization, the average coverage rate of the improved 3D space coverage algorithm was increased by 0.65% and the coverage time increased slightly. Moreover, the proposed method is also used to solve the three-dimensional surface node coverage of the WSN.

**Keywords:** wireless sensor network; 3D space coverage; three-dimensional improved virtual force coverage (3D-IVFC) algorithm; covering method; three-dimensional surface

## 1. Introduction

With the widespread application of the Internet of Things (IoT) technology, sensors play a significant role in wireless sensor networks (WSNs). It is also an important component of the perception layer of the IoT, and the integration, miniaturization, and networking technologies of sensors have gradually matured. A sensor network is usually composed of a huge number of sensor nodes with limited energy [1]. Through mutual cooperation between nodes, it processes the detection data of sensing objects and provides users with accurate and comprehensive real-time data. WSNs have been widely used in military, transportation, environmental monitoring, and other fields [2].

The three-dimensional deployment research of WSNs (see Figure 1) mainly includes underwater sensor networks (UWSNs) [3], underground sensor networks (UGSNs) [4], spatial three-dimensional deployment [5,6], and three-dimensional surface coverage [7,8]. The three-dimensional coverage research on WSNs mainly includes node coverage [5], area coverage [9], and barrier coverage [10]. Among them, the three-dimensional coverage issue is one of the key problems in the WSN research field [11], which reflects the perceived service quality provided by the sensor network and is also a significant indicator for evaluating the performance of the sensor network.

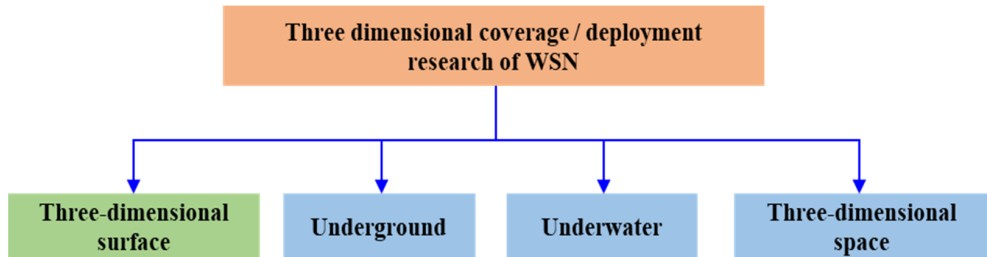

**Figure 1.** Three-dimensional node coverage/deployment problem.

For the study of spatial three-dimensional coverage, Ammari et al. [12] applied the Reuleaux tetrahedral model to sensor node deployment in three-dimensional space, which improved k-fold coverage and connectivity between nodes. Zhong et al. [13] studied the deployment, coverage, and connection of three-dimensional sensor networks, and analyzed the deployment of multiple regular polyhedrons in three-dimensional space. Boukerche et al. [14] proposed a connectivity-based full coverage method for solving two-dimensional and three-dimensional coverage issues in WSN. For research on the 3D coverage control algorithm, Liu et al. [15] proposed a 3D space redeployment method for sensor networks via combined virtual force. Li et al. [16] proposed a 3D space autonomous deployment algorithm based on virtual force compensation. Tang et al. [17] proposed a three-dimensional mobile sensor network autonomous deployment algorithm via a Voronoi diagram to improve the network coverage rate of the monitored area. Chen et al. [18] proposed a coverage control algorithm based on virtual potential field and learning automata, which effectively improved the coverage of directed sensor networks.

The above-mentioned studies are all about the coverage of three-dimensional sensor networks, which are carried out from the directions of 3D coverage, 3D coverage control algorithms, and 3D deployment of directed sensor nodes. However, most of the study approaches consisted of theoretical modeling and simulation. In particular, the virtual force algorithm (VFA) [19] was first proposed in 2003, and it was mainly applied to the two-dimensional node deployment and positioning problems of WSNs. Therefore, the existing advanced research on the VFA are mainly biased towards the node deployment or coverage of WSNs in two-dimensional areas, including node coverage optimization [20,21], mobile node coverage [22], obstacle coverage optimization [23,24], and hybrid swarm intelligence algorithms of coverage optimization [25,26]. However, the study applied to three-dimensional node coverage [27] has the disadvantages of insufficient parameter setting analysis and insufficient experimental comparison.

To solve the three-dimensional space coverage of WSNs, a three-dimensional improved virtual force coverage (3D-IVFC) technique is proposed in this study. In summary, the main contributions and highlights of this article are:

(1) The 3D space coverage problem is theoretically modeled and analyzed, and four 3D space coverage strategies are discussed, namely, full space coverage (FSC), tangent method coverage (TMC), quadrilateral coverage (QC), and theoretical volume coverage (TVC).

(2) Based on the modeled and analyzed 3D space coverage problem, a virtual force coverage algorithm in three-dimensional space is proposed, and the virtual force parameters are demonstrated. In addition, an adaptive adjustment strategy for parameters of the improved algorithm is proposed.

(3) The performance of the 3D-IVFC method is verified by the following experiments, which are different initial coverage strategies (random and centered).

We describe the three-dimensional space coverage problem of the WSN, and an improved three-dimensional virtual force coverage algorithm is also proposed in this paper. Section 2 dives into the three-dimensional coverage of WSNs and discuss four three-dimensional space coverage strategies, namely, full space coverage, tangent method coverage, quadrilateral coverage, and theoretical volume coverage. Section 3 introduces the virtual force coverage algorithm in three-dimensional space and proposes an adaptive

adjustment strategy to control the parameters of the proposed approach in the optimization coverage process. A simulation experiment is carried out to explain the performance of the 3D-IVFC approach in Section 4. Section 5 contains the discussion of the proposed method. The experimental conclusions and prospects are given in Section 6.

## 2. Coverage Problem in Three-Dimensional Space

### 2.1. Node Perception Model

In the real world, the sensing range of a sensor is generally in all directions—that is, an ideal sphere—and the node is the center of the sensing sphere. The connection between sensors is also an indispensable issue in a sensor network. The ideal three-dimensional perception model of the sensor node is shown in Figure 2.

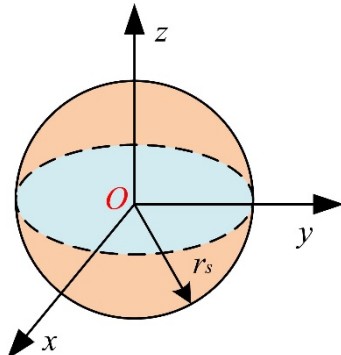

**Figure 2.** 3D sensor node perception model.

In Figure 2, $O$ represents the theoretical center of the sensor node. $r_s$ denotes the sensing radius of the senser node—that is, the Euclidean distance between the point to be monitored—and the theoretical center is shorter than the sensing radius, which will be covered. Then, the volumetric perception range of the node can be calculated as:

$$V = \frac{4\pi r_s^3}{3} \tag{1}$$

### 2.2. Three-Dimensional Point Coverage

The problem of three-dimensional point coverage in WSNs is modeled in this study. It is assumed that there are $n$ detection points to be covered in three-dimensional space, and the sensing radius of sensors is the same. Let the sensing radius be $r_s$ and the communication radius be $r_c$, with the unit of meter (m), where $2r_s \leq r_c$. Assuming that there are $n$ target points in the space to be monitored, if the position coordinates of the $i$-th target point to be monitored are $(x_i, y_i, z_i)$, the position coordinates of sensor node $s$ are $(x_s, y_s, z_s)$. Hence, the spatial Euclidean distance that the sensor can cover of the target to be monitored can be expressed as:

$$d(i,s) = \sqrt{(x_s - x_i)^2 + (y_s - y_i)^2 + (z_s - z_i)^2} \tag{2}$$

Assuming that the probability that the target node $i$ to be monitored is perceived by the sensor node $s$ as $p$, then

$$p(i,s) = \begin{cases} 0, & if\ d(i,s) > r_s \\ 1, & otherwise \end{cases} \tag{3}$$

where $p$ denotes the probability that the target node is covered by the sensor. $d$ is the Euclidean distance between target point and coverage sensor.

### 2.3. Three-Dimensional Coverage Rate

By modeling the target points in three-dimensional space, it can be analyzed from a geometric point of view. The geometric diagram of node coverage in three-dimensional space is shown in Figure 3.

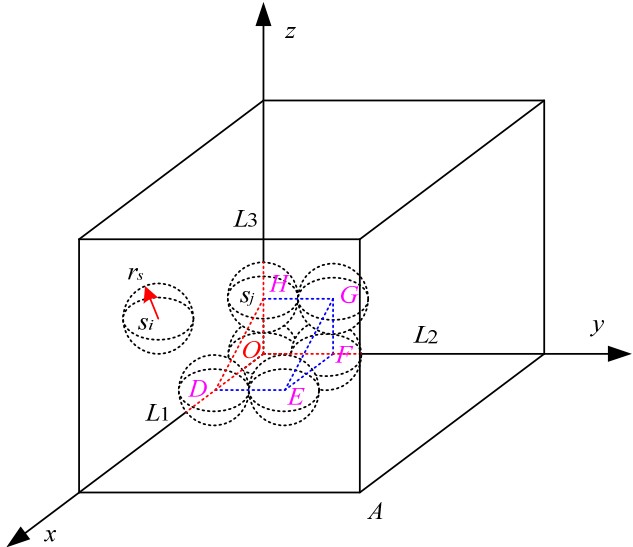

**Figure 3.** Schematic diagram of 3D node coverage.

The schematic diagram is covered by the nodes in Figure 3 and projected onto the two-dimensional plane of *xoy*, as shown in Figure 4. According to the direct spatial Euclidean distance of nodes, it can be divided into three cases: spatial full coverage, tangent coverage, and quadrilateral coverage.

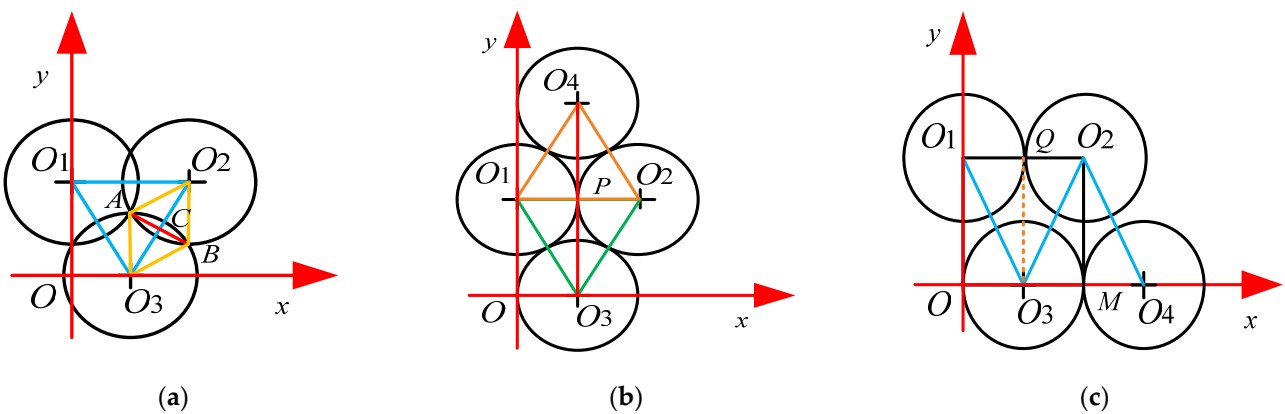

(**a**)    (**b**)    (**c**)

**Figure 4.** Coverage method analysis in a 2D plane of *xoy*. (**a**) Full space coverage. (**b**) Tangent coverage. (**c**) Quadrilateral coverage.

As shown in Figure 4a, the node can achieve full coverage of the spatial area. $O_1$, $O_2$, and $O_3$ represent the positions of the three nodes. At this time, the triangle $O_1O_2O_3$ is an equilateral triangle; $O_3A = r$, that is, the sensing radius of the sensor; and $\angle AO_3B = \pi/3$, $AB = BO_3 = O_3A = r$. According to the nature of the circle, $AB \perp O_2O_3$, $\angle AO_3C = 1/2\angle AO_3B = \pi/6$. According to the cosine theorem, the length of segment $O_3C$ is:

$$L_{O_3C} = L_{O_3A} \times \cos(\angle AO_3C) = r \times \cos(\pi/6) = \frac{\sqrt{3}}{2}r \qquad (4)$$

Then, the length of segment $O_2O_3$ between nodes $O_2$ and $O_3$ is:

$$L_{O_2O_3} = 2L_{O_3C} = \sqrt{3}r \tag{5}$$

As shown in Figure 4b, if nodes are deployed tangentially in three-dimensional space, coverage blind areas will be generated. It can be seen from the figure that the triangle $O_1O_2O_3$ is an equilateral triangle, that is, $O_1P = r$, $\angle O_1O_3O_2 = \pi/3$, $O_1O_3 = O_3O_2 = O_2O_1 = 2r$. According to the properties of regular triangle $O_4O_3 \perp O_2O_1$, $\angle O_1O_3P = 1/2\angle O_1O_3O_2 = \pi/6$. According to the cosine theorem, the length of line segment $O_3P$ is:

$$L_{O_3P} = L_{O_1O_3} \times \cos(\angle O_1O_3P) = \sqrt{3}r \tag{6}$$

Then, the length of segment $O_3O_4$ between nodes $O_3$ and $O_4$ is:

$$L_{O_3O_4} = 2L_{O_3P} = 2\sqrt{3}r \tag{7}$$

For the quadrilateral coverage method shown in Figure 4c, the coverage blind area generated by this method will be slightly larger than that of the tangent method. In addition, the quadrilateral $O_1OO_4O_2$ is a square and the side length is $2r$.

According to the above three coverage methods and the calculation method of ideal volume coverage, the number of nodes required in theory is:

$$N = \frac{V_C}{V} = \frac{V_C}{4\pi r^3/3} \tag{8}$$

$$M = \left(\frac{L}{\sqrt{3}/2 \cdot r + r} + 1\right)^3 \tag{9}$$

$$K = \left(\frac{L}{2r} + 1\right)^2 \cdot \left(\frac{L}{\sqrt{3}/2 \cdot r + r} + 1\right) \tag{10}$$

$$P = \left(\frac{L}{2r}\right)^3 \tag{11}$$

In Equations (8)–(11), $V_C$ represents the volume of deployment space. $N$ is the number of nodes required to deploy the strategy based on the theoretical volume. $M$ denotes the number of nodes by the full coverage method in the theoretical space. $K$ is the number of nodes required by the tangent coverage in theory. $P$ denotes the number of nodes of quadrilateral deployment coverage in theory. $r$ represents the sensing radius of the node. $L$ is the range of three dimensions of the three-dimensional space coverage area.

Under a different sensing radius, the range of node coverage area is set to 500 m × 500 m × 500 m. As the simulation result, the number of nodes required for the corresponding theoretical full coverage is shown in Figure 5.

It can be seen from Figure 5 that the spatial full coverage method requires the largest number of nodes, which can achieve a good monitoring effect, but at a high cost. Using the tangent method, the number of nodes required is less than the theoretical volume coverage and spatial full coverage. Therefore, in the actual deployment application of a WSN, the same quadrilateral deployment coverage can be used as an alternative theoretical design scheme on the premise of meeting the needs of regional monitoring so as to reduce the cost.

Assuming that the three-dimensional space area is divided to be deployed into step $q$ along the $x$, $y$, and $z$ axes, then the length of each section is $l = q$ and the intersection of the deployment space is $q^3$. According to Equations (2) and (3), the node coverage rate (Cov) can be calculated as:

$$\text{Cov} = \frac{p_{\text{cov}}}{q^3} \tag{12}$$

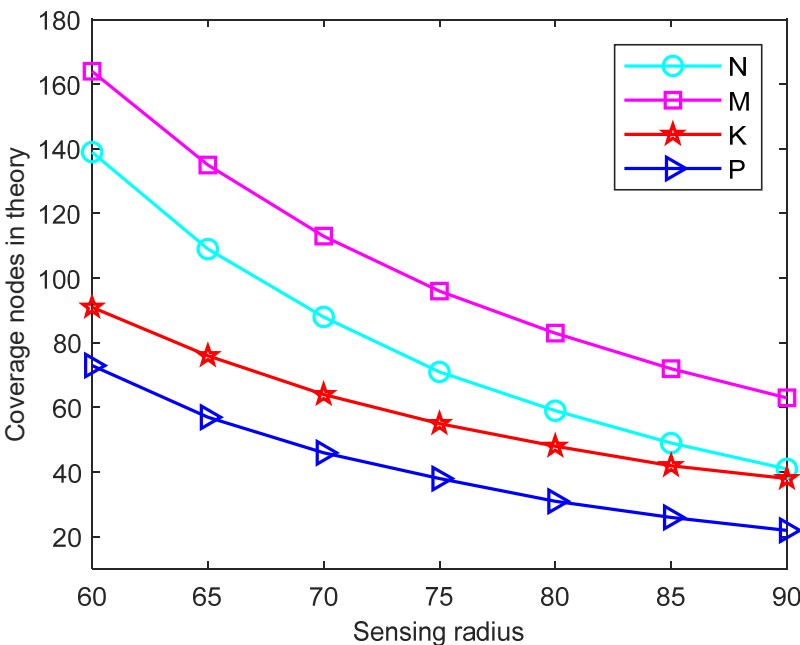

**Figure 5.** Coverage method analysis in theory.

## 3. Three-Dimensional Improved Virtual Force Coverage (3D-IVFC) Algorithm

### 3.1. Node Initial Position

In three-dimensional space, assuming that the initialization position of sensor nodes adopts the method of random deployment, its expression can be defined as:

$$
\begin{cases}
X_i = (X_{\max} - X_{\min}) \cdot rand(n, 1) + X_{\min} \\
Y_i = (Y_{\max} - Y_{\min}) \cdot rand(n, 1) + Y_{\min} \\
Z_i = (Z_{\max} - Z_{\min}) \cdot rand(n, 1) + Z_{\min}
\end{cases}
\tag{13}
$$

where $(X_i, Y_i, Z_i)$ is the spatial position of the node. $n$ is the deployment nodes ($i$ = 1, 2, 3, ..., $n$). $X_{min}$, $Y_{min}$, and $Z_{min}$ denote the lower bounds of the deployment space. $X_{max}$, $Y_{max}$, and $Z_{max}$ represent the upper bounds of the deployment space. *rand* denotes a random number matrix between (0, 1).

### 3.2. Virtual Force Resultant Force

The resultant force between nodes in 3D space is expressed as follows:

$$
F_i = \sum_{j=1, j \neq m}^{m} F_{ij} + F_{ib}
\tag{14}
$$

where $m$ is the number of neighbor nodes of node $s_i$. $F_{ij}$ denotes the virtual resultant force of the neighbor node. $F_{ib}$ represents the virtual resultant force in the boundary area.

#### 3.2.1. Force between Nodes

In node coverage, there are mutual forces between different nodes according to their physical properties, namely, gravity and repulsion. The expression is as follows:

$$
F_{ij} =
\begin{cases}
(\omega_a(D - d_{ij}), \alpha_{ij} + \pi), & if \ d_{ij} < D \\
0, & if \ d_{ij} = D \\
(\omega_b(d_{ij} - D), \alpha_{ij}), & if \ D < d_{ij} \leq r_c \\
0, & if \ d_{ij} > r_c
\end{cases}
\tag{15}
$$

where $F_{ij}$ represents the interaction force between the *i*-th node and *j*-th node in three-dimensional space. $\omega_a$ and $\omega_b$ represent the repulsive force and gravitational coefficient of the virtual force between nodes, respectively. $\alpha_{ij}$ denotes the azimuth between the *i*-th node and *j*-th node. $D$ denotes the distance threshold between nodes. $r_c$ is the communication radius of the node.

### 3.2.2. Area Boundary Limitation

At the boundary of the three-dimensional deployment area, the Euclidean distance $d_{ib}$ between the node and area boundary is shorter than the safe distance threshold. The resultant force formula is as follows:

$$F_{ib} = \begin{cases} (\omega_{a1}(D_b - d_{ib}), \alpha_{ib} + \pi), & if\ d_{ib} < D_b \\ 0, & otherwise \end{cases} \tag{16}$$

where $D_b$ denotes the safety distance threshold between the node and boundary, and $D_b = D/2$. $\omega_{a1}$ denotes the repulsion coefficient of the virtual force between the node and region boundary. $\alpha_{ib}$ represents the azimuth between node *i* and the region boundary. Then the region boundary resultant force of the node in space includes three dimensions: *x*, *y*, and *z*. The formula is:

$$F_{ib} = \sum_{k=1}^{3} F_{ib}^x + \sum_{k=1}^{3} F_{ib}^y + \sum_{k=1}^{3} F_{ib}^z \tag{17}$$

### 3.3. Node Mobility Strategy

In three-dimensional space, under the action of virtual resultant force $F_i$, the sensor node will move from position $P(x_{i\_old}, y_{i\_old}, z_{i\_old})$ to target position $P_1(x_{i\_new}, y_{i\_new}, z_{i\_new})$. The calculation formula of node movement is as follows:

$$x_{i\_new} = \begin{cases} x_{i\_old}, & if\ |F_{xyz}| = 0 \\ x_{i\_old} + \frac{F_x}{F_{xyz}} \times Dis_{max} \times e^{-\frac{1}{F_{xyz}}}, & otherwise \end{cases} \tag{18}$$

$$y_{i\_new} = \begin{cases} y_{i\_old}, & if\ |F_{xyz}| = 0 \\ y_{i\_old} + \frac{F_y}{F_{xyz}} \times Dis_{max} \times e^{-\frac{1}{F_{xyz}}}, & otherwise \end{cases} \tag{19}$$

$$z_{i\_new} = \begin{cases} z_{i\_old}, & if\ |F_{xyz}| = 0 \\ z_{i\_old} + \frac{F_z}{F_{xyz}} \times Dis_{max} \times e^{-\frac{1}{F_{xyz}}}, & otherwise \end{cases} \tag{20}$$

where $F_x$, $F_y$, and $F_z$ represent the projection of virtual force of nodes in three-dimensional space on the *x*-axis, *y*-axis, and *z*-axis, respectively. $F_{xyz}$ denotes the virtual force resultant force on the node. $Dis_{max}$ is the maximum distance the node moves each time.

The average moving distance of the node is one of the indicators to judge the effectiveness of the node coverage of the algorithm in the three-dimensional space area, and its calculation formula is:

$$\bar{d} = \frac{1}{n}\sum_{i=1}^{n} \sqrt{(x_i' - x_i)^2 + (y_i' - y_i)^2 + (z_i' - z_i)^2} \tag{21}$$

where $\bar{d}$ denotes the average moving distance of the node. *n* is the number of deployment nodes. $(x_i, y_i, z_i)$ indicates the initial position of the sensor node. $(x_i', y_i', z_i')$ represents the moved position of the node.

### 3.4. Adaptive Virtual Force Parameters $\omega_a, \omega_b$

The classical VFA was first proposed for solving the two-dimensional node location problem. This paper extends the research of the virtual force algorithm, which promotes the two-dimensional coordinates to three-dimensional coordinates, and we applied it to handle the three-dimensional node coverage issue. The virtual force parameters $\omega_a$ and

$\omega_b$ of the 3D coverage problem were analyzed, and the schematic diagram is presented in Figure 6.

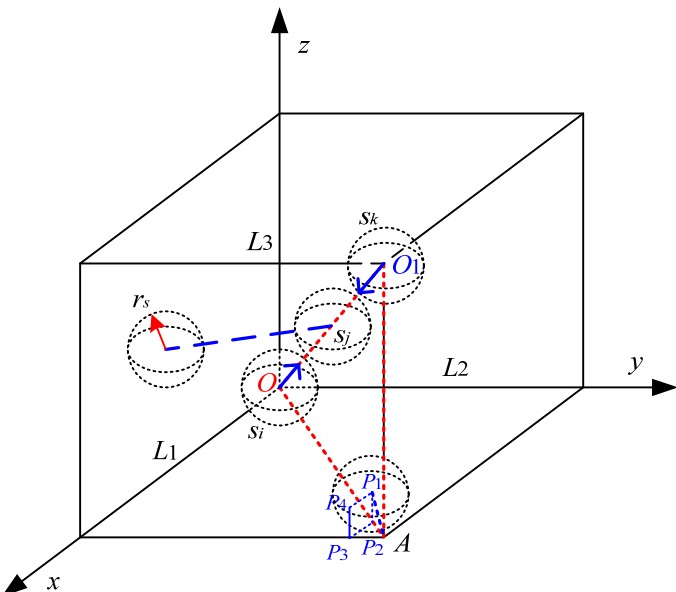

**Figure 6.** Virtual force analysis of end nodes.

During the deployment of mobile nodes, the repulsion parameter between nodes is much larger than the gravity parameter. Suppose that the length (*x*-axis), width (*y*-axis), and height (*z*-axis) of the spatial area are $L_1$, $L_2$, and $L_3$, respectively; the three end nodes are $O$, $O_1$, and $P_1$, respectively; the position of node $s_j$ is located on the connecting line between end nodes $O$ and $O_1$; and the projection point $P_2$ of node $P_1$ (tangent to the three edge interfaces of the space) is located on the line segment $OA$. According to the above assumptions and geometric principles, the quadrilateral $P_1 P_2 P_3 P_4$ is a square and its side length is the perceived radius length ($r_s$) of the point, here $AP_3 = r_s$, so,

$$AP_1 = \sqrt{(P_1 P_2)^2 + (P_2 P_3)^2 + (AP_3)^2} = \sqrt{3} r_s \tag{22}$$

In the three-dimensional space coverage area, all deployment points will be affected by the gravitational force between nodes, so the gravitational force on any end node is:

$$F_j^a = (n - 2) \cdot \omega_a \cdot \left( \sqrt{L_1^2 + L_2^2 + L_3^2} - \sqrt{2}(r_i + r_k) \right) \tag{23}$$

Its repulsive force is:

$$F_j^b = \omega_b \cdot (D_b - d_{jk}) \tag{24}$$

In Equation (23), $r_i$ and $r_k$ represent the Euclidean distance projected between the center point of the end node and each boundary, respectively, and its value range is $(0, \sqrt{3} r_s)$. $n$ indicates the number of deployment nodes. In Equation (24), $d_{jk}$ is the spatial Euclidean distance between node $s_j$ located near the end node and the end node $s_k$.

When the repulsion and gravity of the nodes in the deployment area are balanced, that is, $F_j^a = F_j^b$, the following can be obtained:

$$(n - 2) \cdot \omega_a \cdot \left( \sqrt{L_1^2 + L_2^2 + L_3^2} - \sqrt{2}(r_i + r_k) \right) = \omega_b \cdot (D_b - d_{jk}) \tag{25}$$

In Equation (25), when the deployment node $n$ is much greater than 2 and $\sqrt{L_1^2 + L_2^2 + L_3^2}$ is much greater than $\sqrt{2}(r_i + r_k)$, it can be simplified as:

$$n \cdot \omega_a \cdot \sqrt{L_1^2 + L_2^2 + L_3^2} = \omega_b \cdot (D_b - d_{jk}) \tag{26}$$

That is,

$$\begin{cases} \omega_a = (D_b - d_{jk}) \\ \omega_b = n \cdot \sqrt{L_1^2 + L_2^2 + L_3^2} \end{cases} \tag{27}$$

In the experimental design, the value of repulsion $\omega_b$ is generally much greater than that of gravity $\omega_a$, which conforms to Equation (26).

### 3.5. Complexity Analysis

For the above analysis, the pseudo-code of the 3D space node deployment strategy via the improved virtual force coverage approach is as Algorithm 1:

---

**Algorithm 1:** Pseudo-code of the 3D-IVFC algorithm.

---

Input: Monitoring area 500 m × 500 m × 500 m, the coordinate position of the node, and the maximum iterations Max_iter.
1. Set the scope of the three-dimensional space area: $X_{max}$ = 500, $X_{min}$ = 10; $Y_{max}$ = 500, $Y_{min}$ = 10; $Z_{max}$ = 500, $Z_{min}$ = 10, and the step length of the point to be monitored is 25.
2. Randomly deploy the initial position of nodes, and calculate coverage and unmonitored points $k$.
3. For t = 1: Max_iter
4.　　For $i$ = 1: $n$
5.　　　For $k$ = 1: $k$
6.　　　　Calculate the distance between the node and the unmonitored point using Equation (2).
7.　　　　If dik > $r_s$
8.　　　　　Update the parameters $\omega_a, \omega_b$ using Equation (26).
9.　　　　　Calculate the resultant force $F$ using Equations (13), (15)–(17).
10.　　　　End
11.　　　End
12.　　　Move the node in space using Equations (18)–(20).
13.　　End
14.　　For $i$ = 1: $n$
15.　　　For $j$ = 1: $n$
16.　　　Calculate the distance between nodes using Equation (2).
17.　　　If dik > $r_s$ and dik < = $r_c$
18.　　　　Calculate the resultant force $F$ using Equations (13), (15)–(17).
19.　　　End
20.　　End
21.　　Update the position of the node in space using Equations (18)–(20).
22.　　Determine whether the node location exceeds the deployed space.
23. End
24. Update the coverage.
25. End
Output: Optimized coverage, node location, and node movement trajectory

---

It can be seen from the above pseudo-code that the complexity of the deployment algorithm proposed in this research mainly includes node location initialization $O(n)$, movement of nodes and regional boundaries and unmonitored points $O(n \times k)$, and movement between nodes $O(n^2)$. Then, the complexity of the proposed method is:

$$O(3D - VFA) = O(n) + O(t) \times O\left(n \times k + n^2\right) \tag{28}$$

## 4. Simulation Results

### 4.1. Experimental Environment and Parameter Setting

To analyze the effectiveness of the algorithm proposed in this study, two groups of tests were designed: (1) When the sensing radius and the number of deployment nodes were certain, the effects of the 3D-VFC algorithm and the 3D-IVFC algorithm on node space coverage under random initial deployment were studied. (2) Additionally, the performance of the two comparison algorithms on node 3D space coverage under centralized random

initial deployment was studied. The experimental environment of this study was on Windows 10, Intel Core i5-10210U CPU @2.11G with 8G RAM, Matlab 2018a.

For the above four node coverage strategies, the node coverage in three-dimensional space of the simulation area was realized with the proposed algorithm. In the simulation experiment in this paper, the initial position of nodes was randomly deployed, and the parameter settings of the proposed algorithm are shown in Table 1. One of the improved algorithms [21] of the VFA was used to compared with our method. Based on the theoretical analysis, the number of coverage nodes in the simulation test was set to 63 by the FSC model in this paper. In addition, each group was tested 10 times independently to confirm the persuasiveness of the experimental results.

**Table 1.** Parameter setting.

| Parameters | 3D-VFC | IVFA | 3D-IVFC |
|---|---|---|---|
| Deployment area side length/m | $500 \times 500 \times 500$ | $500 \times 500 \times 500$ | $500 \times 500 \times 500$ |
| Number of nodes | 63 | 63 | 63 |
| Perceived radius $r_s$/m | 90 | 90 | 90 |
| Communication radius/m | $r_c = 2r_s$ | $r_c = 2r_s$ | $r_c = 2r_s$ |
| Max_iter | 30 | 30 | 30 |
| Repulsion and gravitational coefficient | $\omega_a = 1, \omega_b = 5$ | $\omega_a = 1, \omega_b = 1000$ | Calculated by Equation (26) |
| Maximum boundary moving step/m | 5 | 5 | 5 |
| Maximum node moving step/m | 10 | 10 | 10 |
| Boundary safety distance threshold/m | $\sqrt{2}r_s$ | $\sqrt{3}r_s$ | $\sqrt{2}r_s$ |

### 4.2. Different Initial Deployment Strategies

To verify the effectiveness of the proposed method, the comparison simulations were carried out under different initial deployment conditions. For case 1, the initial nodes were deployed randomly. However, the initial nodes were deployed in the middle area for case 2. The simulation results of two cases of the 3D-IVFC algorithm are presented in Figures 7 and 8. The statistical results were taken into consideration and each group of experiments was tested 10 times independently. The comparison results of the experiment are shown in Tables 2 and 3. Additionally, the boxplots show the superiority and stability of the performance of the proposed technique.

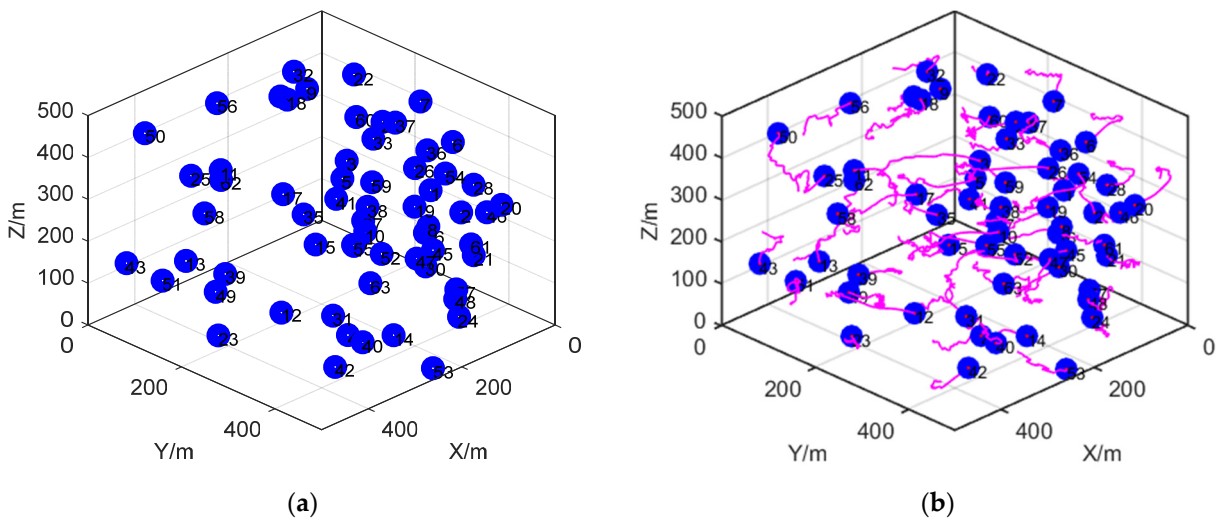

(**a**)                                                                                        (**b**)

**Figure 7.** *Cont.*

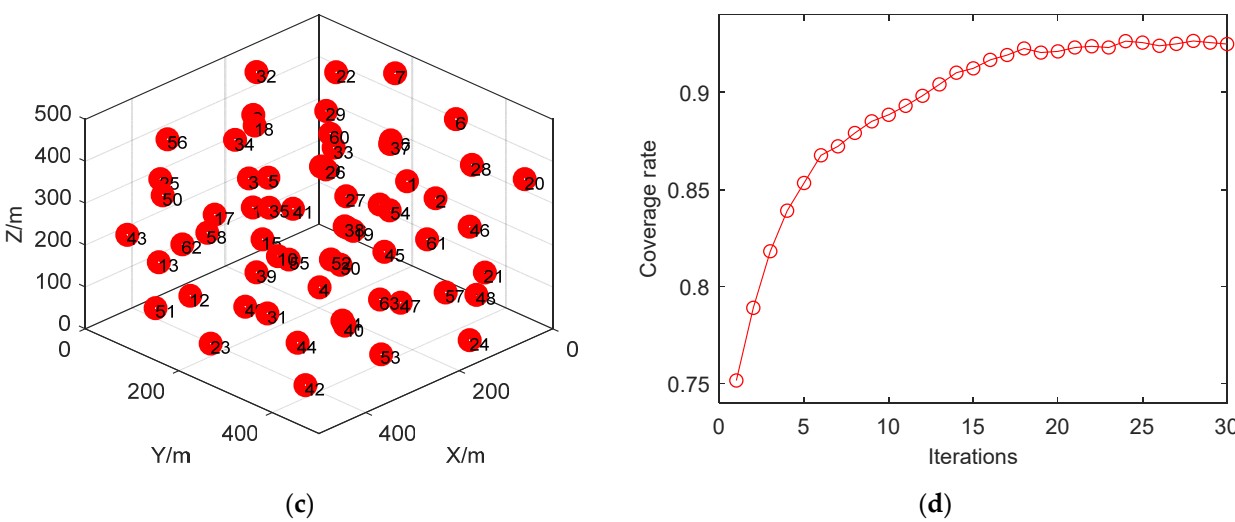

(**c**)   (**d**)

**Figure 7.** Random 3D space node movement process. (**a**) Node initial deployment. (**b**) Node movement track. (**c**) Optimized node deployment. (**d**) Coverage curve.

(**a**)   (**b**)

(**c**)   (**d**)

**Figure 8.** 3D space node movement process via centered. (**a**) Node initial deployment. (**b**) Node movement track. (**c**) Optimized node deployment. (**d**) Coverage curve.

**Table 2.** Algorithm comparison of random node coverage results.

| Algorithm | No. | 1 | 2 | 3 | 4 | 5 | 6 | 7 | 8 | 9 | 10 | Average |
|---|---|---|---|---|---|---|---|---|---|---|---|---|
| 3D-VFC | Initialization (%) | 74.76 | 72.41 | 74.83 | 74.30 | 72.64 | 73.18 | 76.21 | 79.39 | 76.81 | 76.43 | 75.10 |
|  | Optimal (%) | 90.15 | 91.69 | 91.29 | 90.90 | 91.75 | 91.26 | 92.00 | 91.06 | 91.88 | 91.90 | 91.39 |
|  | Time (s) | 3.97 | 2.58 | 2.54 | 2.74 | 2.50 | 2.50 | 2.76 | 2.69 | 2.74 | 2.54 | 2.76 |
| IVFA | Initialization (%) | 75.26 | 74.16 | 73.59 | 70.95 | 75.53 | 74.56 | 74.58 | 72.78 | 73.10 | 70.99 | 73.55 |
|  | Optimal (%) | 91.04 | 91.36 | 91.90 | 91.34 | 92.05 | 91.20 | 91.40 | 91.58 | 91.55 | 91.65 | 91.51 |
|  | Time (s) | 3.01 | 2.87 | 2.98 | 2.78 | 2.80 | 3.09 | 4.86 | 2.73 | 3.86 | 2.84 | 3.18 |
| 3D-IVFC | Initialization (%) | 77.59 | 74.93 | 74.40 | 77.38 | 72.93 | 76.64 | 72.83 | 72.38 | 75.90 | 72.95 | 74.79 |
|  | Optimal (%) | 92.48 | 92.05 | 92.18 | 92.36 | 92.06 | 91.91 | 92.34 | 91.85 | 92.15 | 92.09 | 92.15 |
|  | Time (s) | 2.89 | 2.50 | 2.49 | 2.86 | 2.61 | 2.73 | 2.40 | 2.47 | 2.44 | 2.46 | 2.59 |

**Table 3.** Comparison of algorithms for coverage results of central nodes.

| Algorithm | No. | 1 | 2 | 3 | 4 | 5 | 6 | 7 | 8 | 9 | 10 | Average |
|---|---|---|---|---|---|---|---|---|---|---|---|---|
| 3D-VFC | Initialization (%) | 37.19 | 37.45 | 36.89 | 39.11 | 40.03 | 39.66 | 38.81 | 36.69 | 38.93 | 40.70 | 38.55 |
|  | Optimal (%) | 91.21 | 91.28 | 91.74 | 91.80 | 91.56 | 91.56 | 91.75 | 91.28 | 92.16 | 91.78 | 91.61 |
|  | Time (s) | 3.05 | 2.98 | 2.69 | 2.43 | 2.61 | 2.53 | 2.54 | 3.07 | 2.96 | 2.73 | 2.76 |
| IVFA | Initialization (%) | 39.39 | 37.08 | 38.73 | 39.58 | 37.30 | 39.46 | 39.96 | 38.41 | 38.59 | 42.01 | 39.05 |
|  | Optimal (%) | 91.76 | 92.05 | 91.55 | 91.99 | 91.93 | 91.33 | 91.99 | 92.15 | 91.11 | 91.50 | 91.74 |
|  | Time (s) | 2.98 | 2.93 | 2.81 | 2.82 | 2.87 | 3.00 | 2.90 | 2.78 | 2.75 | 2.79 | 2.86 |
| 3D-IVFC | Initialization (%) | 38.43 | 38.45 | 40.30 | 40.39 | 36.64 | 36.84 | 38.46 | 37.38 | 41.55 | 36.34 | 38.48 |
|  | Optimal (%) | 92.16 | 92.50 | 92.14 | 92.28 | 92.31 | 92.29 | 92.14 | 92.20 | 92.28 | 92.26 | 92.26 |
|  | Time (s) | 2.83 | 2.93 | 2.61 | 2.87 | 2.89 | 2.91 | 2.64 | 2.95 | 2.68 | 2.60 | 2.79 |

4.2.1. Case 1: Initial Node Random Deployment (3D-IVFC Algorithm)

When nodes are randomly deployed, it can be seen from Figure 7a that there is centralization and overlap, resulting in a poor coverage effect. In Figure 7b, through the redeployment of the nodes by the 3D-IVFC algorithm, the position of the nodes moves, and the curve represents the movement track of the nodes in the algorithm iteration process. In Figure 7c, the nodes are evenly distributed in the monitored spatial area after the deployment of the set iteration times of the algorithm. In Figure 7d, the coverage increases gradually with the increase in the number of iterations.

Based on the 10 separate experiments shown in Table 2, the average coverage rate of the 3D-VFC method was 91.39% and the coverage time was 2.76 s. The average coverage rate obtained with IVFA was 91.51% and the coverage time was 3.18 s. The average coverage rate of the 3D-IVFC algorithm was 92.15% and the deployment time was 2.59 s. The average coverage increased by 0.76% and 0.64%, respectively, and the deployment time decreased by 0.17 s and 0.59 s, respectively.

4.2.2. Case 2: Initial Node Location Centered Deployment (3D-IVFC Algorithm)

It can be seen from Figure 8a that the initial deployment location adopted by the node is concentrated in the middle area of the deployment space. In Figure 8b, the nodes are redeployed through the 3D-IVFC algorithm, and the nodes move from the middle position to surrounding the original position. The curve in the figure represents the movement track of the nodes in the algorithm iteration process. In Figure 8c, after the deployment of the set iteration times of the algorithm, the nodes can be evenly distributed in the monitored spatial area. Additionally, the coverage increases gradually with the increase in the number of iterations in Figure 8d.

Based on the 10 separate experiments shown in Table 3, the average coverage rate of the 3D-VFC algorithm was 91.61% and the deployment time was 2.76 s. The average coverage rate obtained with IVFA was 91.74% and the coverage time was 2.86 s. The average coverage rate of the 3D-IVFC algorithm was 92.26% and the deployment time was

2.79 s. The average coverage increased 0.65% more than the 3D-VFC algorithm, and the deployment time increased slightly. Moreover, the average coverage increased 0.52% more than IVFA, and the deployment time decreased slightly.

The standard deviations of the 3D-VFC algorithm and IVFA in the cases were $5.78 \times 10^{-3}$ and $2.96 \times 10^{-3}$, and $3.07 \times 10^{-1}$ and $3.47 \times 10^{-1}$, respectively, and the standard deviations of the 3D-IVFC algorithm were $2.00 \times 10^{-3}$ and $1.08 \times 10^{-3}$, respectively. It proves the stability of the performance of the proposed approach. Particularly, the proposed 3D-VFC method is superior than the 3D-VFC algorithm and IVFA, as shown in Figure 9.

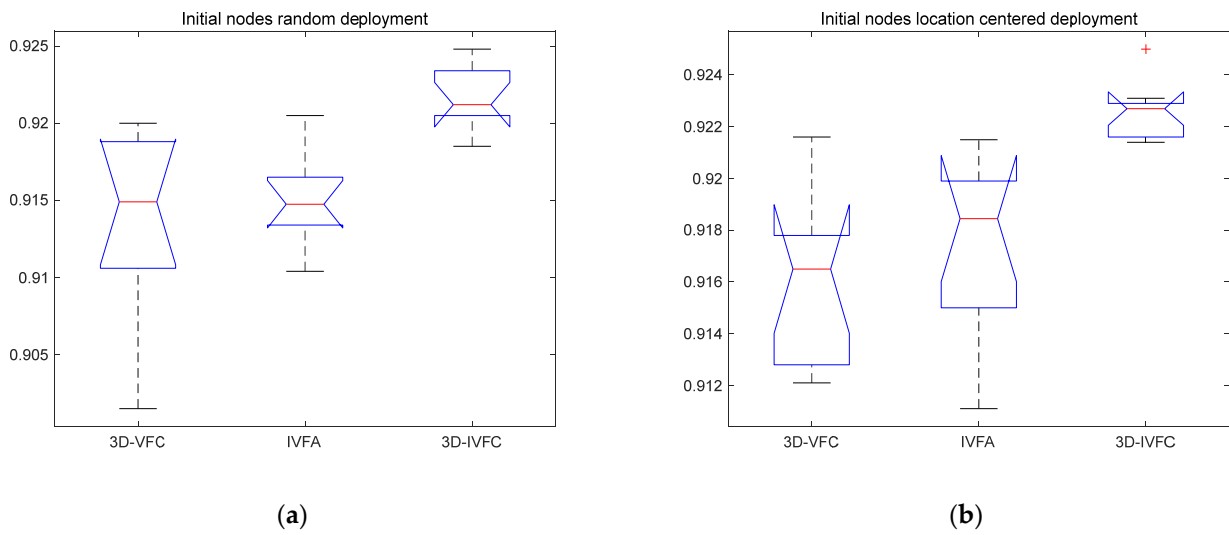

**Figure 9.** The boxplot results of the proposed algorithm in both cases. (**a**) Comparison boxplot results of case 1. (**b**) Comparison boxplot results of case 2.

### 4.2.3. D-VFC Algorithm for the Surface Coverage Issue

Based on the comparison results, the proposed 3D-VFC algorithm can obtain superior performance than 3D-VFC algorithm and IVFA in this paper. Additionally, to verify the validity of the proposed method on the surface coverage issue, we add the simulation experiment on the surface under the space of $1000 \text{ m} \times 1000 \text{ m} \times 100 \text{ m}$. The number of sensor nodes was set to 60 and the sensing radius was $r_s = 90$ m, whereas the communication radius $r_c = 180$ m and the number of iterations was set to 20. The simulation results of the 3D-IVFC algorithm for solving the surface coverage task are presented in Figures 10 and 11.

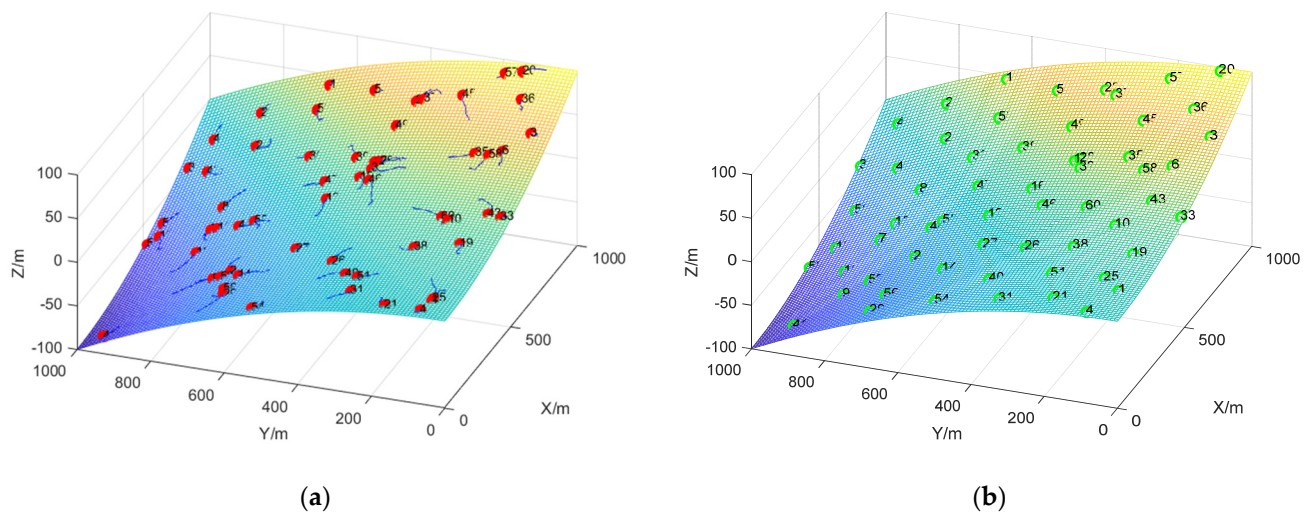

**Figure 10.** 3D surface node movement process via random initialization. (**a**) Initial position and movement track of nodes. (**b**) Optimized node deployment.

The surface function is $z = \left( x^2 + y^2 \right) / h^2$, where $h$ is the height of the three-dimensional space, and $h = 100$ m in the simulation test. As seen from Figures 10 and 11, the 3D-IVFC algorithm had a better application effect on the 3D surface node coverage problem. After 20 iterations, the node coverage in the set coverage area increased from 76.81% to 96.62%, an increase of 19.81%, and the time was 8.34 s for the three-dimensional surface coverage.

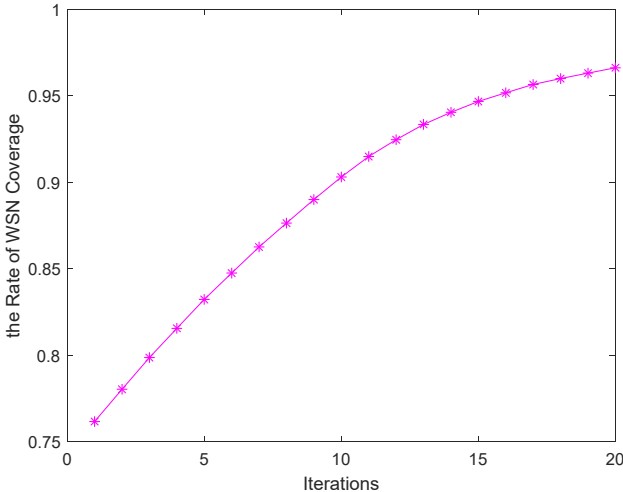

**Figure 11.** Coverage curve of the 3D-IVFC algorithm for the 3D surface issue.

## 5. Discussion

Based on the comparison results in Tables 2 and 3, the average coverage rate of the proposed 3D-IVFC method was 92.15% and the deployment time was 2.56 s with 10 iterations for each experiment with the comparison algorithms. The average coverage increased by 0.76% and 0.64%, and the deployment time decreased by 0.17 s and 0.59 s for the 3D-VFC method and IVFA, respectively, when the initial deployment was random. Moreover, the average coverage rate of the 3D-IVFC approach was 92.26% and the deployment time was 2.79 s. When the initial deployment was centered, the average coverage increased by 0.65% and 0.52% for the 3D-VFC method and IVFA, respectively, and the deployment time increased and decreased slightly, respectively.

As can be seen in Figure 9, the proposed 3D-IVFC algorithm is more robust than the 3D-VFC algorithm and IVFA and has superior performance. In addition, the proposed method was also used to solve the three-dimensional surface coverage issue. However, the performance of the 3D-IVFC algorithm can be further improved in future work, not only for the three-dimensional space deployment tasks but also for three-dimensional surface coverage and communication problems.

## 6. Conclusions

We explored a novel 3D-IVFC algorithm via the basic virtual force algorithm in this paper, which was used to solve a three-dimensional space coverage problem. Firstly, the deployment strategies of four node numbers in three-dimensional space were analyzed, and the number of nodes required for space full coverage, tangent coverage, quadrilateral coverage, and theoretical volume coverage were compared and discussed. Secondly, for four cases, the number of nodes required for setting 3D space coverage was calculated theoretically, and the three-dimensional space deployment was simulated and tested by the 3D-IVFC algorithm. Finally, the full coverage of the three-dimensional space was verified by simulation experiments. Future work will carry out the following research: (1) The 3D-IVFC approach proposed in this study can be used for a directed sensor coverage problem in three-dimensional space. (2) The swarm intelligence optimization algorithm [28–30] can also be combined with the 3D-IVFC algorithm and can be applied to solve the coverage and positioning of three-dimensional directional sensors in WSNs. (3) The distributed

deployment of heterogeneous networks will be studied in combination with Voronoi partition strategy and swarm intelligence optimization, as well as the establishment of multi-objective problems such as WSN coverage, energy, and network lifetime [31,32]. (4) In the distributed deployment task using the wireless sensors [20], which can improve the capability of safety and numerical control for the management, the proposed method may also be useful and effective in future research.

**Author Contributions:** Methodology, M.Z.; writing—original draft preparation, M.Z. and J.Y.; supervision, T.Q.; funding acquisition, J.Y. All authors have read and agreed to the published version of the manuscript.

**Funding:** This work was supported by the NNSF of China (No. 61640014), the Industrial Project of Guizhou Province (No. Qiankehe Zhicheng [2022]017, [2019]2152), the Innovation Group of the Guizhou Education Department under Grant Qianjiaohe (No. KY [2021]012), the Science and Technology Fund of Guizhou Province under Grant Qiankehe (No. [2020]1Y266), Qiankehejichu (No. ZK [2022]Yiban103), the Science and Technology Foundation of Guizhou University (Guidateganghezi [2021]04), CASE Library of IoT (KCALK201708), and the platform about IoT from Guiyang National High-Tech Industrial Development Zone (No. 2015).

**Institutional Review Board Statement:** Not applicable.

**Informed Consent Statement:** Not applicable.

**Data Availability Statement:** Not applicable.

**Conflicts of Interest:** The authors declare no conflict of interest.

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
