# Peer review of "An Adaptive Three-Dimensional Improved Virtual Force Coverage Algorithm for Nodes in WSN"

_axioms, doi:10.3390/axioms11050199_

Round 1
Reviewer 1 Report
Parameters omega "a" and "b" are mentioned in the abstract and the introduction. However, the reader has no notion about their meaning because no model has been mathematically described in such sections.
The radius rs is not described in Figure 2, considering this plot as a sphere.
In line 101, "perception range" can be changed to "volumetric perception range".
Include some spaces in Eq. (3) to improve the format. Why this probability was considered binary, radially, the perception (high-probability) is improved in the center of the sphere.
Equations (15) and (16) should be reformated to be clearly interpreted.
Equations (18), (19), and (20) use "if-else" conditionals, i.e., very different style regarding the previous equations. Please check and revise.
In Table 2, please change "/%"by " [%]", "/s" by "[s]". and so on.
Xlabel in Figure 8(d) should be "Iterations".
In the results and discussion sections, authors argue improvement using random processes, but the number of repetitions to obtain the Statistiques was not mentioned.
Results should include more comparisons with other state-of-the-art methods.
The bibliographical references should be updated, choosing relevant journal contributions from the last five years.
Author Response
We appreciate the reviewer for his/her precious and valuable comments that help us improve the quality of the paper. In the revised paper and the response letter, the mentioned issues have been fully considered and addressed.
Comments and Suggestions for Authors
- Parameters omega "a" and "b" are mentioned in the abstract and the introduction. However, the reader has no notion about their meaning because no model has been mathematically described in such sections.
Response:
For the reader to understand what meaning of the parameters ωa and ωb, we have redescribed the parameters in the abstract and the introduction of the revised manuscript.
- The radius rs is not described in Figure 2, considering this plot as a sphere.
Response:
In the revised manuscript, the radius rs has been described in Figure 2.
- In line 101, "perception range" can be changed to "volumetric perception range".
Response:
In line 105 of the revised manuscript, we have changed "perception range" by "volumetric perception range".
- Include some spaces in Eq. (3) to improve the format. Why this probability was considered binary, radially, the perception (high-probability) is improved in the center of the sphere.
Response:
In Eq. (3) of the revised manuscript, we have reformatted ithe revised manuscript. The reason for binary probability used in this paper that it is easy to make reader understand the node coverage issue in theory, and simulate by the research. In future work, we will take the energy and communication of the nodes in the WSN into consideration.
- Equations (15) and (16) should be reformated to be clearly interpreted.
Response:
In Equations (15) and (16), we have reformatted in the revised manuscript.
- Equations (18), (19), and (20) use "if-else" conditionals, i.e., very different style regarding the previous equations. Please check and revise.
Response:
We have checked and revised the Eq. (18) to Eq. (20) of the revised manuscript.
- In Table 2, please change "/%"by " [%]", "/s" by "[s]". and so on.
Response:
In Tables 2 and 3 of the revised manuscript, we have changed the "/%" by " [%]", and "/s" by "[s]".
- Xlabel in Figure 8(d) should be "Iterations".
Response:
In Figures 8(d) and 9(d), we have updated the X-label by “Iterations” in the revised manuscript.
- In the results and discussion sections, authors argue improvement using random processes, but the number of repetitions to obtain the Statistiques was not mentioned.
Response:
In 310, 328, 360and 361, we have added the number of independent experiments in the results and discussion sections in the revised manuscript.
- Results should include more comparisons with other state-of-the-art methods.
Response:
We have taken the improvement of basic virtual force algorithm (VFA) into consideration and chosen one of the improved VFA in recent years based on the two-dimensional VFA. The relatively fair comparison experiments have been added in the revised manuscript, which the reference is as follows:
[1] Hu, T., Zhong, S. Research on a virtual force algorithm in Wireless Sensor Network. Proc. of the International Conference on Frontiers of Electronics, Information and Computation Technologies, 2021, 1-5.
Additionally, we have added a simulation of the 3D-VFC algorithm for solving the three-dimensional surface issue in WSN. The experiment results are shown as follows:
From the comparison results, the proposed 3D-VFC algorithm can obtain superior performance than 3D-VFC algorithm and IVFA in this paper. Additionally, to verify the validity of the proposed method on surface coverage issue, we add the simulation experiment on the surface under the space of 1000 × 1000 × 100 m. The number of sensor nodes is set to 60, and the sensing radius is = 90 m, communication radius = 180 m, and the number of iterations is 20. The simulation results of 3D-IVFC algorithm for solving the surface coverage problem are presented in Fig. 10 and Fig. 11.
(a) Initial position and movement track of nodes |
(b) Optimized node deployment |
Figure 10. 3D surface node movement process via random initialization
Figure 11. Coverage curve of 3D-IVFC algorithm on 3D surface issue
The surface function is, where the h is the height of the three-dimensional space, that is h=100m in the simulation test. As seen in Fig. 10 and Fig. 11, 3D-IVFC algorithm has a better application effect on 3D surface node coverage problem. After 20 iterations, the node coverage in the set coverage area increased from 76.81% to 96.62%, an increase of 19.81%, and the time-consuming is 8.34 s of the three-dimensional surface coverage.
- The bibliographical references should be updated, choosing relevant journal contributions from the last five years.
Response:
As suggested, we have added descriptions of related work of the references in the revised manuscript. The updated references are as follows:
[1] Zafer, M., Senouci, M. R., and Aissani, M. Efficient deployment approach of wireless sensor networks on 3D terrains. International Journal of Data Mining, Modelling and Management, 2021, 13, 114-136.
[2] Si, P., Ma, J., Tao, F., Fu, Z., and Shu, L. Energy-efficient barrier coverage with probabilistic sensors in wireless sensor networks. IEEE Sensors Journal, 2020, 20, 5624-5633.
[3] Saad, A., Senouci, M. R., and Benyattou, O. Toward a realistic approach for the deployment of 3D Wireless Sensor Networks. IEEE Transactions on Mobile Computing, 2022, 21, 1508-1519.
[4] Boukerche, A., Sun, P. Connectivity and coverage-based protocols for wireless sensor networks. Ad Hoc Networks, 2018, 80, 54-69.
[5] Hu, T., Zhong, S. Research on a virtual force algorithm in Wireless Sensor Network. Proc. of the International Conference on Frontiers of Electronics, Information and Computation Technologies, 2021, 1-5.
[6] Ji, K., Zhang, Q., Yuan, Z., Cheng, H., Yu, D. A virtual force interaction scheme for multi-robot environment monitoring. Robotics and Autonomous Systems, 2022, 149, 103967.
[7] Yao, Y., Li, Y., Xie, D., Hu, S., Wang, C., Li, Y. Coverage enhancement strategy for WSNs based on virtual force-directed ant lion optimization algorithm. IEEE Sensors Journal, 2021, 21, 19611-19622.
[8] Singh, A., Sharma, S., and Singh, J. Nature-inspired algorithms for wireless sensor networks: A comprehensive survey. Computer Science Review, 2021, 39, 100342.
[9] Liu, X., Wang, X., Jia, J., and Huang, M. A distributed deployment algorithm for communication coverage in wireless robotic networks. Journal of Network and Computer Applications, 2021, 180, 103019.
***************************************************************************
Thanks a lot for the invaluable comments and suggestions. The comments from the editors and reviewers are significant, informative, and constructive, which greatly help us improve the quality of the manuscript.

Reviewer 2 Report
The authors present a timely and interesting paper. The proposed method is presented in detail.
The reviewer believes that only smaller modifications are required before publication. Such as the number of references. It is adequate, although the reviewer believes that a few more should be added to strengthen the scientific background of the paper.
The English used is good, and the text only requires a minor spellcheck.
Author Response
Thanks a lot for the invaluable comments and suggestions, these suggestions are very valuable for our paper, and we carefully revise the manuscript according to these suggestions.
Comments and Suggestions for Authors
- The authors present a timely and interesting paper. The proposed method is presented in detail.
Response:
Thank you for your recognition of our paper.
- The reviewer believes that only smaller modifications are required before publication. Such as the number of references. It is adequate, although the reviewer believes that a few more should be added to strengthen the scientific background of the paper.
Response:
As suggested, we have changed these references [1-4] by the last four years of relevant research, such as:
[1] Zafer, M., Senouci, M. R., and Aissani, M. Efficient deployment approach of wireless sensor networks on 3D terrains. Inter-national Journal of Data Mining, Modelling and Management, 2021, 13, 114-136.
[2] Si, P., Ma, J., Tao, F., Fu, Z., and Shu, L. Energy-efficient barrier coverage with probabilistic sensors in wireless sensor networks. IEEE Sensors Journal, 2020, 20, 5624-5633.
[3] Saad, A., Senouci, M. R., and Benyattou, O. Toward a realistic approach for the deployment of 3D Wireless Sensor Networks. IEEE Transactions on Mobile Computing, 2022, 21, 1508-1519.
[4] Boukerche, A., Sun, P. Connectivity and coverage-based protocols for wireless sensor networks. Ad Hoc Networks, 2018, 80, 54-69.
In addition, we have also added several references in the introduction [5-7] and future work [8,9] of the revised manuscript as follows:
[5] Hu, T., Zhong, S. Research on a virtual force algorithm in Wireless Sensor Network. Proc. of the International Conference on Frontiers of Electronics, Information and Computation Technologies, 2021, 1-5.
[6] Ji, K., Zhang, Q., Yuan, Z., Cheng, H., Yu, D. A virtual force interaction scheme for multi-robot environment monitoring. Robotics and Autonomous Systems, 2022, 149, 103967.
[7] Yao, Y., Li, Y., Xie, D., Hu, S., Wang, C., Li, Y. Coverage enhancement strategy for WSNs based on virtual force-directed ant lion optimization algorithm. IEEE Sensors Journal, 2021, 21, 19611-19622.
[8] Singh, A., Sharma, S., and Singh, J. Nature-inspired algorithms for wireless sensor networks: A comprehensive survey. Computer Science Review, 2021, 39, 100342.
[9] Liu, X., Wang, X., Jia, J., and Huang, M. A distributed deployment algorithm for communication coverage in wireless robotic networks. Journal of Network and Computer Applications, 2021, 180, 103019.
- The English used is good, and the text only requires a minor spellcheck.
Response:
As suggested, we have corrected the mentioned typos and carefully checked the rest of the revised manuscript.
***************************************************************************
Thanks a lot for the invaluable comments and suggestions. The comments from the editors and reviewers are significant, informative, and constructive, which greatly help us improve the quality of the manuscript.

Round 2
Reviewer 1 Report
The authors have addressed all my concerns.
Some minimal issues related to the last modifications should be carefully checked, for instance, "1000 × 1000 × 100m" should be m^3 ?.
I recommend a deep-proofread for acceptance.
Author Response
We appreciate the reviewer for his/her precious and valuable comments that help us improve the quality of the paper. In the revised paper and the response letter, the mentioned issues have been fully considered and addressed.
In line 348 of the revised manuscript, we have checked and revised the "m" by "m^3".
Additionally, as suggested, we have corrected the mentioned typos and carefully checked the rest of the revised manuscript.
Thanks a lot for the invaluable comments and suggestions. The comments from the editors and reviewers are significant, informative, and constructive, which greatly help us improve the quality of the manuscript.